# Coherence transfer from optically induced THz magnons to charges

Moritz Cimander[1], Volker Wiechert [1], Julian Bär [1], Takuya Satoh [2], Jörg Bünemann[3], Götz S. Uhrig [3] & Davide Bossini [1] ✉

The digital economy, society and politics are increasingly shaped by cloud-based data. The advancement of transformative technologies, such as artificial intelligence, is placing unprecedented demands on data centres. This has driven an intense pursuit of a concept for data storage, manipulation and transfer able to operate even at THz rates while minimizing energy dissipation. Collective spin excitations, namely magnons, have been proposed as energy-efficient information carriers. A critical challenge concerns the integration of this approach with the ubiquitous CMOS technology. This step requires a mechanism to convert THz coherent magnons into a charge signal. Here we demonstrate the coherence transfer from optically driven THz magnons to charges in terms of an optical response. We identify the conditions necessary for this effect and formulate a microscopic model reproducing the experimental results without any fine-tuning of the parameters. These findings offer a pathway toward an energy-efficient, high-speed information technology.

Realizing a concept for information technology, able to outperform present-day schemes while being compatible with charge-based CMOS, is the main drive of the research field of spintronics. The demands for higher operational frequency and lower energy dissipation motivate the identification of magnons as information carriers, as their generation and detection do not necessarily involve Joule heating[1,2]. Further following this line of reasoning, antiferromagnets (AF) have emerged as a promising material platform, in view of their intrinsically high magnon frequencies, entering the THz regime[3–7]. These eigenfrequencies correspond to (sub)-picosecond timescales, which can be accessed by means of femtosecond laser pulses. Optical methods have been employed to drive, manipulate and detect coherent THz magnons, both at the centre[4,5,8] and at the edges[9–11] of the Brillouin zone. The grand challenge of the magnon-to-charge conversion has ignited scientific interest in the coupling between spin and charge dynamics on the (sub)-picosecond timescale in AF(s). Several concepts have been investigated such as light-driven magnetic transport through AF/heavy metal interfaces[12–18], coupling of GHz magnons with electrons photoinduced in the conduction band and excitons in van der Waals semiconductors[19,20]. A different approach, based on the optical pumping of a composite excitation known as exciton-

magnon[21,22], has also been explored. This strategy has revealed coupled dynamics of electrons and magnons[23], photo-induced phase transition[24] and nonlinear magnonic dynamics, wherein different magnon modes are coupled in the transient state[25,26]. Despite this remarkable volume of research, a general concept to convert coherent magnons with THz frequencies into a charge signal in a bulk dielectric AF is lacking. We stress the relevance of considering bulk dielectric AFs, which represent by far the majority of materials magnetically ordered in nature. These systems lack the symmetry requirements for magnetoelectricity and multiferroicity, which on the other hand couple magnetism to charges both in the ground[27,28] and excited[29] states.

Here, we therefore conduct our experiment on the prototypical dielectric AF: NiO. We drive coherent THz magnons with femtosecond laser pulses with photon energy below the bandgap. We probe the electronic response by monitoring the transient transmissivity. Similar experiments revealed that charges and magnons do not couple at the picosecond timescale[5]. A technical development of our apparatus, combined with a systematic investigation disclose the conditions enabling the coherence transfer from photoinduced THz magnons to charges, which results in a modulation of the transient transmissivity. We formulate a microscopic theoretical model addressing the effect of

[1]Department of Physics and Center for Applied Photonics, University of Konstanz, Konstanz, Germany. [2]Department of Physics, Institute of Science Tokyo, Tokyo, Japan. [3]Condensed Matter Theory, TU Dortmund University, Dortmund, Germany. ✉e-mail: davide.bossini@uni-konstanz.de

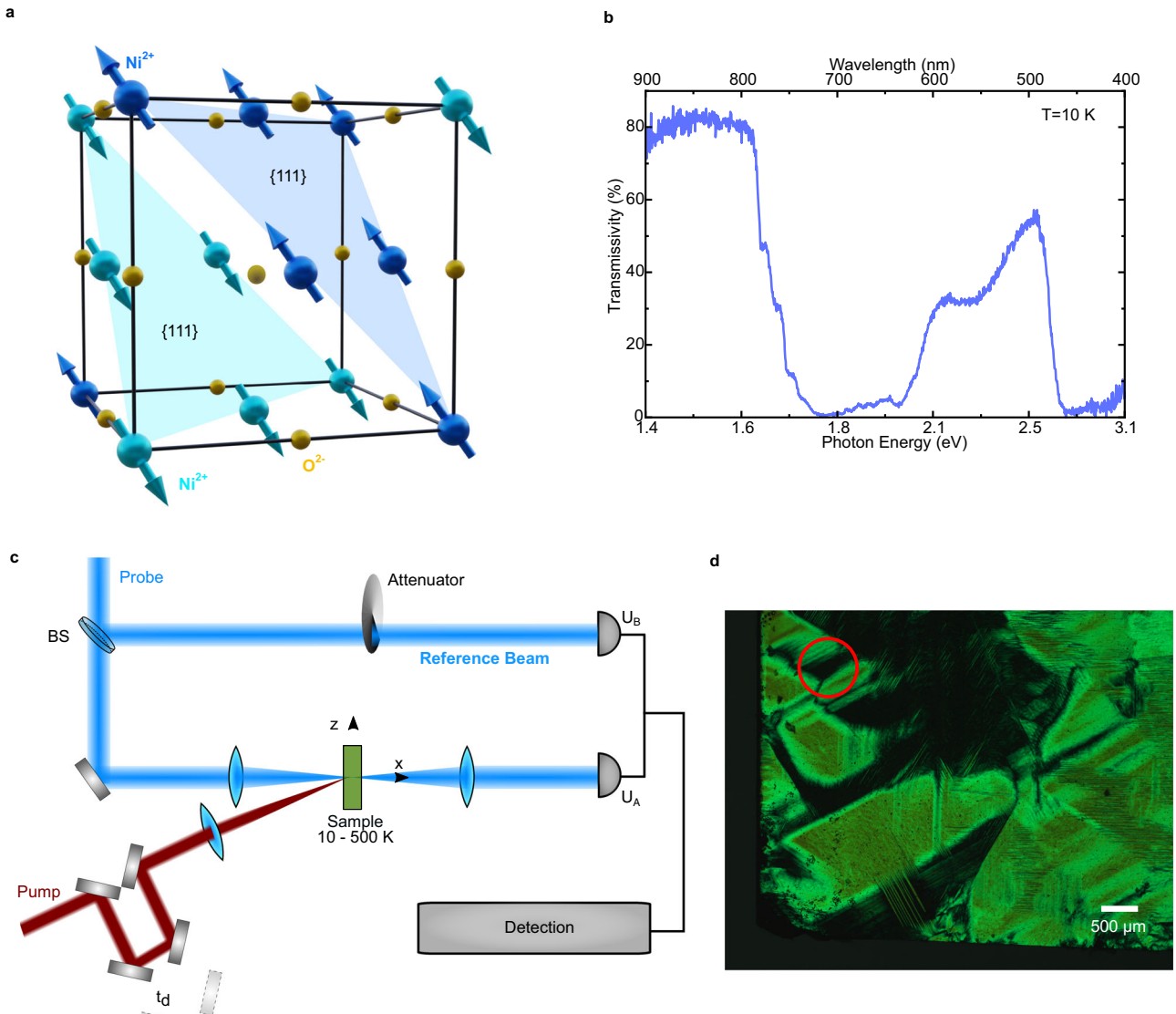

**Fig. 1 | Ground state properties of NiO and experimental approach. a** Unit cell of NiO. The blue arrows indicate the spins of the Ni$^{2+}$ ions aligned in ferromagnetic layers. **b** Static transmittance of a 50 μm thick single crystal of NiO in the VIS-NIR range, measured setting the sample temperature to 10 K. **c** Schematic representation of the experimental set-up. The pump pulses excite the sample, the probe beam interacts with the sample and is then recollected and detected in a photodiode (U$_A$). A separate photodiode detects the intensity of a reference beam (U$_B$).

The difference between the voltages generated by these two photodiodes reveals the transmittance of the sample in a balanced detection scheme (details in the "Methods" section). **d** Microscopy picture of the sample showing the T-domains taken in a Cross-Nicol geometry following the procedure reported in ref. 43. The red circle indicates the T$_0$ domain on which the pump-probe experiments are performed. The white scale bar corresponds to 500 μm.

the generation of magnons onto the optical spectrum of NiO. Specifically, the model predicts a spin-orbit mediated modulation of the energy of several electronic transitions, which generates a transient transmissivity consistent with the observations.

Our specimen is a free-standing single crystal, cut along the **111** plane, with thickness of approximately 50 μm. Below its Néel temperature ($T_N = 523$ K) NiO displays a collinear antiferromagnetic order. Two sublattices of Ni$^{2+}$ spins align ferromagnetically along the $\langle 11\bar{2}\rangle$ axes in {111} planes, with antiferromagnetic coupling between adjacent {111} planes[5] (see Fig. 1a). Figure 1b displays the transmissivity of NiO in the visible and near-infrared ranges measured at $T = 10$ K (details of the measurements in the Methods section). NiO is a charge-transfer insulator with a 4 eV bandgap. The intragap optical transitions visible in Fig. 1b are ascribed to the *d-d* transitions of the Ni$^{2+}$ ($3d^8$) electrons[5,30]. These transitions become weakly electric-dipole allowed, because of

the assistance of the spin-orbit coupling[31,32]. The time-resolved measurements of the transmissivity were performed in the pump-probe scheme (Fig. 1c). The photon-energy of the excitation beam is tuned to 0.98 eV, resonant with a transition to a composite exciton-magnon. The literature reports that driving this process resonantly generates efficiently coherent THz magnons[25]. We vary the photon energy of the probe beam within the range 1.6 eV–2.6 eV for our purposes. While the transient transmissivity can be detected with a single diode, the sensitivity of the set-up can be increased by more than one order of magnitude by balancing the detection (see Methods section and Fig. 1c). The temperature of the sample can be changed in the 5−300 K range. For our investigation, we consider it favorable to study a homogeneous magnetic texture. We thus select experimentally a single T$_0$ domain region of NiO (Fig. 1d), in which all spins lie in the sample plane prior to the photoexcitation.

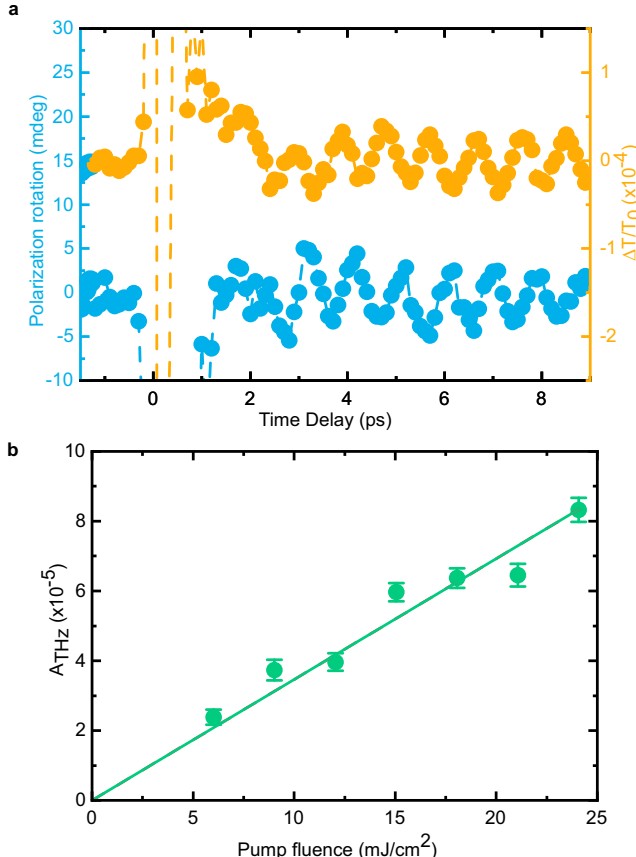

**Fig. 2 | Coherent modulation of the transmissivity via optically induced THz magnons. a** Time traces showing the rotation of the probe polarisation (blue) and the transient transmissivity (orange) photo-induced by the pump beam. The pump photon energy is 0.98 eV in both datasets. The pump pulses are linearly polarized along the horizontal axis. The fluence was set to 24 mJ/cm². The photon energy of the probe pulses is 1.65 eV and they are linearly s-polarized (along the y-axis in the coordinate systems in Fig. 1c) for both data sets. **b** Pump fluence dependence of the oscillations of $\Delta T/T$. The experimental parameters are the same as those employed for the pump-probe traces in Fig. 2a. The drawn line serves as a guide to the eye. The error bars are given by the standard error of the amplitudes obtained by the fitting the data in the time-domain. The standard error is calculated from the width of 95% confidence interval.

## Results

Figure 2a reports the photoinduced dynamics of the transient transmissivity ($\Delta T/T$) observed at 5 K. The signal displays oscillations at the frequency of 1.07 THz, which matches the frequency of one of the magnon modes at the center of the Brillouin zone in NiO[4,5]. Although our pumping scheme has already been demonstrated to be able to generate THz magnons[5,25,33], we perform also a transient magneto-optical measurement (see "Methods" for the magneto-optical detection) under the same experimental conditions. Oscillations at the 1.07 THz frequency are observed in the magneto-optical signal as well, consistently with the well-established results in the literature[5,25,33] (Fig. 2a). In the framework of a local mean-field description the generation of coherent magnon from the zone center corresponds to a slight tilt of the sublattice magnetization (see "Model" in the Method section). Figure 2a reports a surprising result, since similar experiments have not disclosed a coherent THz modulation of the transient transmissivity hitherto[5]. We ascribe this discrepancy to the balanced detection scheme, which effectively improves the sensitivity of our set-up by more than one order of the magnitude (see "Methods"). The THz contribution to the signal is the focus of our work, so we isolate this component in the time traces shown here (see "Methods"). The

temperature dependence of the signal (Supplementary Fig. 1) further confirms the magnetic nature of the THz oscillations. The amplitude of the coherent contribution scales linearly as the intensity of the excitation beam is increased (Fig. 2b). This is consistent with the general description of the exciton-magnon transition in terms of a linear absorption process[21,22,25].

Exploring the dependence of the $\Delta T/T$ signal on the probe photon energy reveals an unexpected and puzzling phenomenon. Figure 3a displays that most of the time-traces, but critically not all of them, are modulated by THz oscillations. This surprising evidence demands to correlate the data in Fig. 3a with the transmissivity spectrum of NiO. In Fig. 3b the spectral content of the probe pulses employed for the time-traces in Fig. 3a is shown, together with the transmissivity. As a first observation, we can exclude the scenario wherein the amplitude of the whole spectrum is modulated by the photoinduced magnons. In this case, coherent oscillations should be detectable at all the photon energies employed for the probe beam in our experiment, which is at odds with the results. Closely inspecting Fig. 3b, we realize that coherent oscillations of $\Delta T/T$ are observed only in spectral ranges where the transmissivity is not flat and thus displays a significant slope. This becomes more obvious in Fig. 3c-d: Within the bandwidth of the probe pulses centered at 2.38 eV and 2.52 eV, light experiences a flat optical response of NiO. This fact suggests that the detected signal originates from the modulation of the energy of one or more $d$-$d$ transitions in the spectrum. This picture naturally explains why coherent THz contributions are not observed in the transient transmissivity of flat spectral regions.

## Discussion

We turn now to the question raised by the data in Fig. 3a: Do the oscillations of the transient transmissivity correspond to a coherence transfer from spins to charges at THz frequency? Fig. 2a and the literature (*11, 31, 38*) demonstrate that magneto-optical effects reveal coherent THz magnons in NiO. We must therefore assess whether and, in a positive case, how magneto-optics affects the transmissivity detected in our experiment and shown in Fig. 3a. To the best of our knowledge, this discussion has not been reported in the literature yet. We systematically analyze the four main linear magneto-optical effects (see Methods), obtaining that: i) the Faraday and Cotton-Mouton effects do not affect the transmissivity. These effects are in essence magnetic contributions to circular and linear birefringence, which thus modify exclusively the polarization of a linearly polarized light beam, but not its transmitted intensity; ii) magnetic circular dichroism induces a change of the transmissivity, independent of the polarization of light; iii) magnetic linear dichroism modifies periodically the transmissivity, as a function of the polarization of the incoming beam ($1 + \cos(2\varphi)$ where $\varphi$ describes the polarization of the beam, see Fig. 4). It follows from this analysis, that the dependence of the coherent THz oscillations on the polarization of the probe beam is key to assess a possible contribution of magneto-optics to our data. Figures 5–9 report that the modulation of the transmissivity detected for all probe photon energies does not arise from magneto-optical effects, except for the data obtained for 1.65 eV (Fig. 8), which are consistent with the symmetry of magnetic linear dichroism. We therefore conclude that we did indeed observe a coherent modulation of the transmissivity, due to the optical generation of magnons. Hence THz coherent spin oscillations have been converted into electronic dynamics.

Any phenomenological model is intrinsically unable to predict the magnitude of a specific effect but can only assess whether the material's symmetry allows it. We thus take a different approach, by developing a microscopic model, specific for NiO. We calculate first the energy of the electronic $d$-$d$ transitions, which lie in the spectral range accessed in our experiment. The transition energies are calculated from the Hamiltonian of a single $Ni^{2+}$ ion in the average magnetic

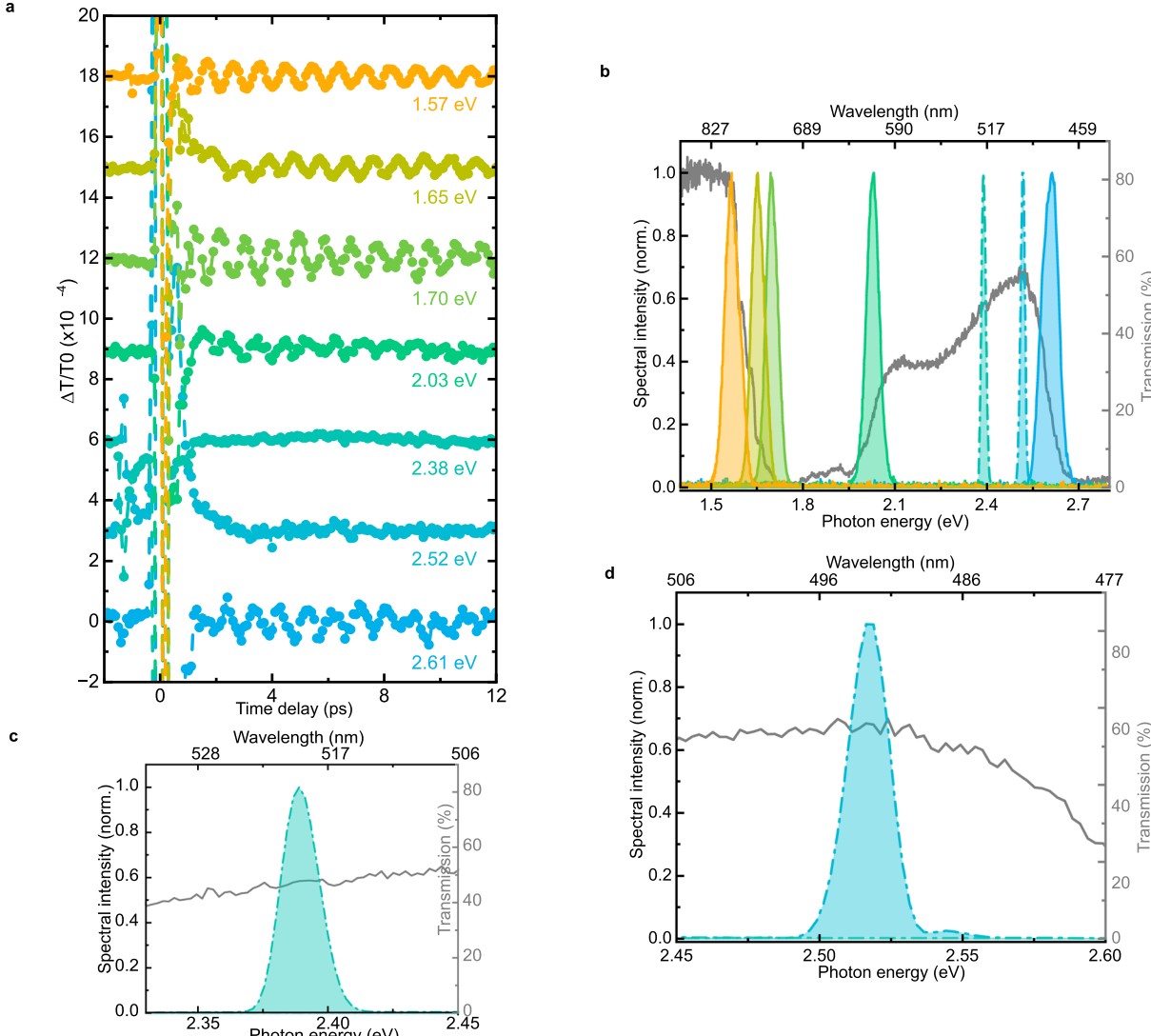

**Fig. 3 | Spectral dependence of the transient transmissivity. a** Transient transmissivity for different probe photon energies between 1.57 eV and 2.61 eV. The pump beam has a photon energy of 0.98 eV, a fluence of 24 mJ/cm² and is linearly p-polarized (0°, parallel to the z axis in Fig. 1c) perpendicular to the probe beam (s-polarized, 90°, parallel to the y axis in Fig. 1c). **b** Static transmissivity of NiO (grey)

and spectrum of the probe beams corresponding to the time traces of (**a**). The dashed lines show the spectrum of the probe beams for which no oscillations are detected. Zoom-in on the 2.38 eV (**c**) and 2.52 eV (**d**) probe spectra, superimposed with the NiO transmissivity (grey line).

field exerted by its neighboring ions, i.e. we employ a magnetic local mean-field approximation. The single-ion Hamiltonian comprises all generic local interactions, crystal field splittings and the spin-orbit coupling while neglecting delocalized band structure effects (details in the Method section). We deem this approximation to be well-grounded for a correlated insulator, in which we induce and detect dynamics of electrons in localized $d$ bands. The energies of these electronic processes lie several electronvolts below the fundamental absorption edge. In other words, no electrons were photoinduced in the conduction band. We start from the unperturbed anti-ferromagnetic ground state (Fig. 10a), for which the $d$-$d$ transitions are represented in Fig. 10b. The outcome of the model (dashed lines) is in good agreement with the spectral features appearing in the experimentally detected transmissivity of NiO. We highlight that the parameters employed were taken from the literature without any fine-tuning (see Methods). In a second step, we calculate how the tilting by angle $\alpha$ of the effective magnetic mean-field, resulting from the optically driven magnons, affects the energy of the $d$-$d$ transitions. The ground state magnetization is tilted from the $\langle 11\bar{2}\rangle$ towards the $\langle 111\rangle$

direction by the angle $\alpha$, consistent with the polarization of the THz magnon mode[33] (see Fig. 10a). The square of the tilting angle is proportional to the population of the coherently pumped zone center magnon in the regime of small $\alpha$ (see Methods). The modification of the energy of the $d$-$d$ transitions as a function of the tilting $\alpha$ is shown in Fig. 10c. Considering the specific value $\alpha = 20$ mrad, the predictions of the model are then compared with the experimental results (details in Methods), displaying a satisfactory agreement (Fig. 10d). The value $\alpha = 20$ mrad is self-consistent with the observations: This limited tilting generates spin dynamics, correctly portrayed in terms of linear response theory. Accordingly, no trace of nonlinearity (e.g. frequency modification, non-linear fluence dependence) is observed in the time-traces reported in our work. In this framework, it is thus not surprising that the energies of the transitions also scale linearly with the tilting. All the experimental evidence combined with the data analysis and modelling confirm that coherence is transferred from the photo-induced THz coherent magnons to charges, modulating as a result the transient transmissivity. We highlight that the key coupling mediating between the magnons and the charge degrees of freedom in our model

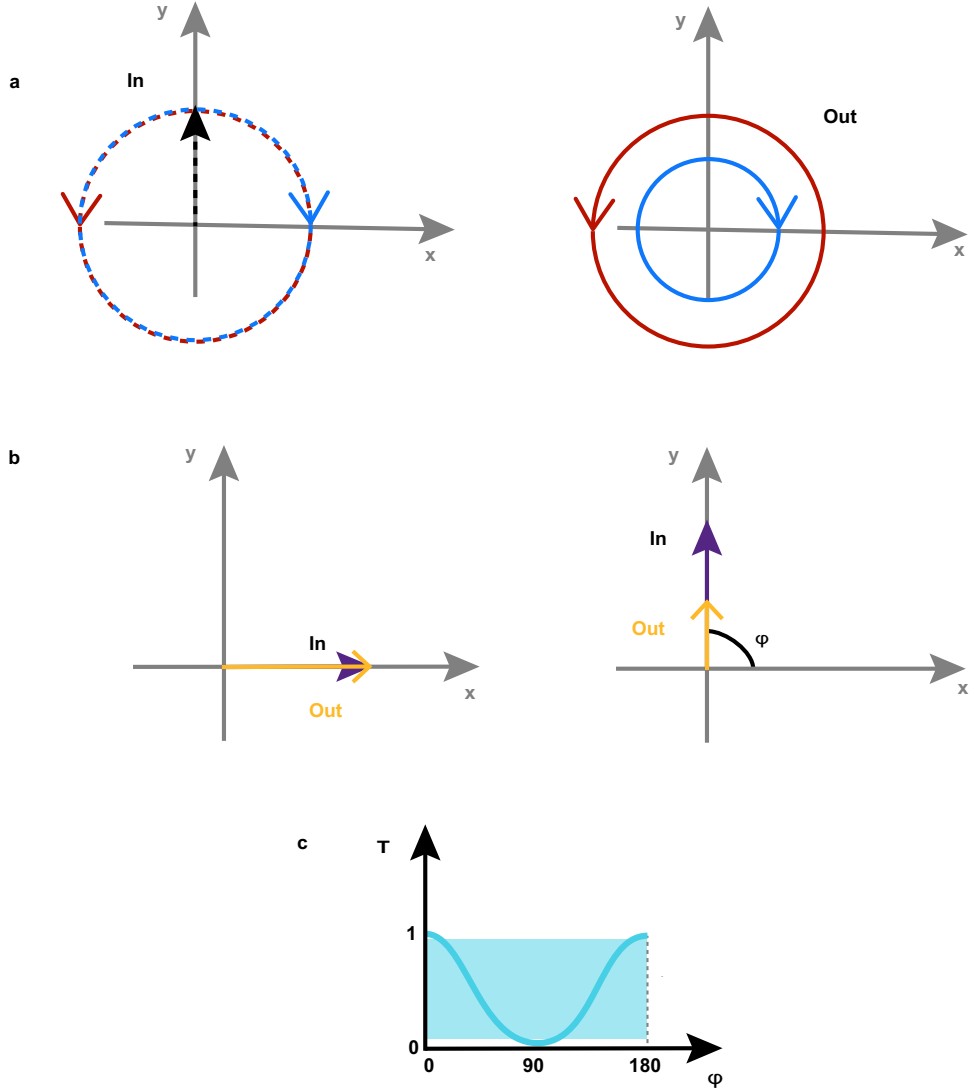

**Fig. 4 | Magneto-optical effects and transmissivity. a** Magnetic circular dichroism. The intensity of an input linearly polarized beam (left) is reduced (right) by a fixed amount, independent of the polarization of the beam. **b** Magnetic linear dichroism. The intensity of an input linearly polarized beam (left) is reduced (right). The effect depends periodically on the polarization of the beam, described by the angle $\varphi$ (right panel). **c** Dependence of the reduction of the intensity due to the magnetic linear dichroism on the angle $\varphi$ in the idealized case (blue line) and real case (blue area) discussed in the Methods section.

is the spin-orbit coupling. This coupling leads to a dependence of the electronic levels on the orientation of the magnetization, i.e. of the effective magnetic field. No magnetostriction nor any other spin anisotropic exchange needs to be invoked. Therefore, strain-mediated effects, which have been employed to interpret the complicated laser-driven spin dynamics of NiO/platinum interfaces[34], are not relevant in our case. As already discussed, the symmetries of the magneto-optical effects are not consistent with our results. Furthermore, we would like to comment on the relation between the canonical symmetry analysis of light-spin interaction, describing magneto-optical effects, and our model. Let us recall that a main approximation underlies the symmetry analysis: The material-specific tensors, which depend on the electronic structure of the system, are constant[35]. The data in Fig. 3 reveal that the energy of electronic transitions of NiO is affected by the optically driven coherent magnons, as supported by our model (Fig. 10c). This result undermines the key assumption of the symmetry analysis: the material-specific tensors – which depend on the electronic structure – are not constant after the illumination, as the photoinduced modification of the *d-d* energy levels affects the electronic environment. Hence, a cohesive and coherent interpretation of the effect disclosed

in our experiment demands to develop the microscopic modelling presented above. The magnon-induced modification of the electronic structure predicted in our model is elusive to the canonical treatment of magneto-optical effects.

Finally, we discuss the role of the magnon excitation mechanism. We relied on the exciton-magnon, as it enables us to resonantly drive the 1.07 THz magnon mode with near-infrared laser pulses. However, the observation of coherent oscillations in the transient transmissivity does not depend on the excitation mechanism, provided that the magnon amplitude is big enough to be detectable. To verify this statement, we compare data obtained by driving magnons via the exciton-magnon transition with the results of impulsive stimulated Raman scattering (Supplementary Fig. 3). Despite having the same incident fluence, the non-resonant excitation provides a less pronounced signal.

Our approach is applicable to several material classes, as d-d transitions allowed by the spin-orbit coupling are commonly observed in a wide range of magnetic dielectrics[36,37]. We foresee that in particular materials with stronger spin-orbit coupling (e.g. CoO and rare-earth orthoferrites) are promising. They could even enable the conversion of

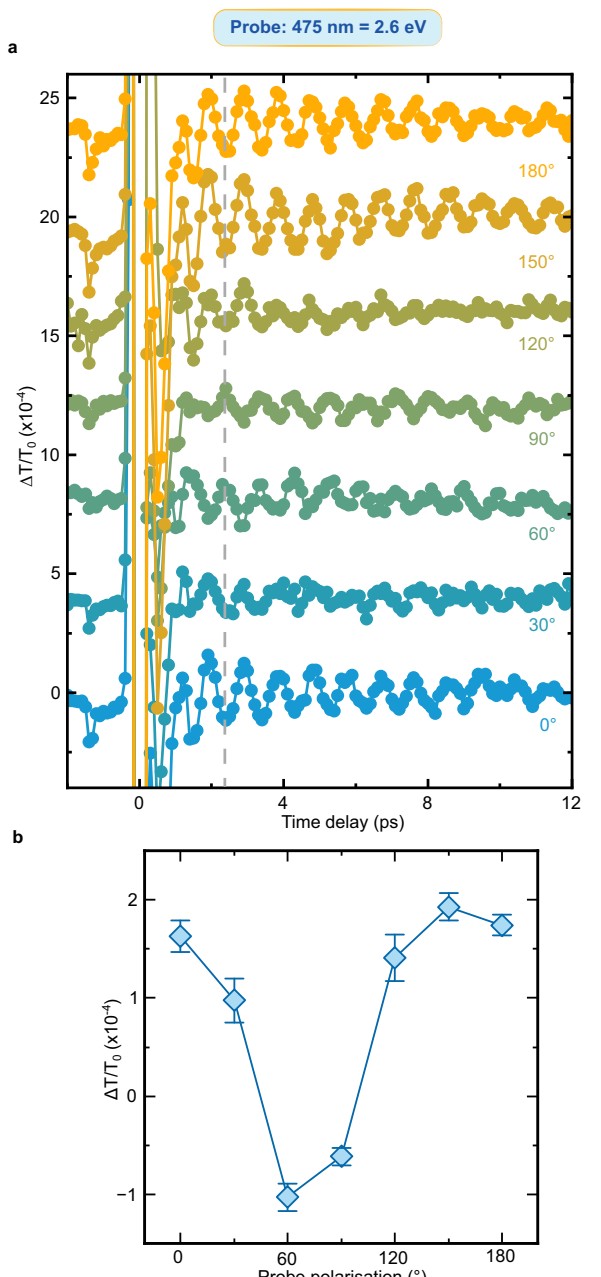

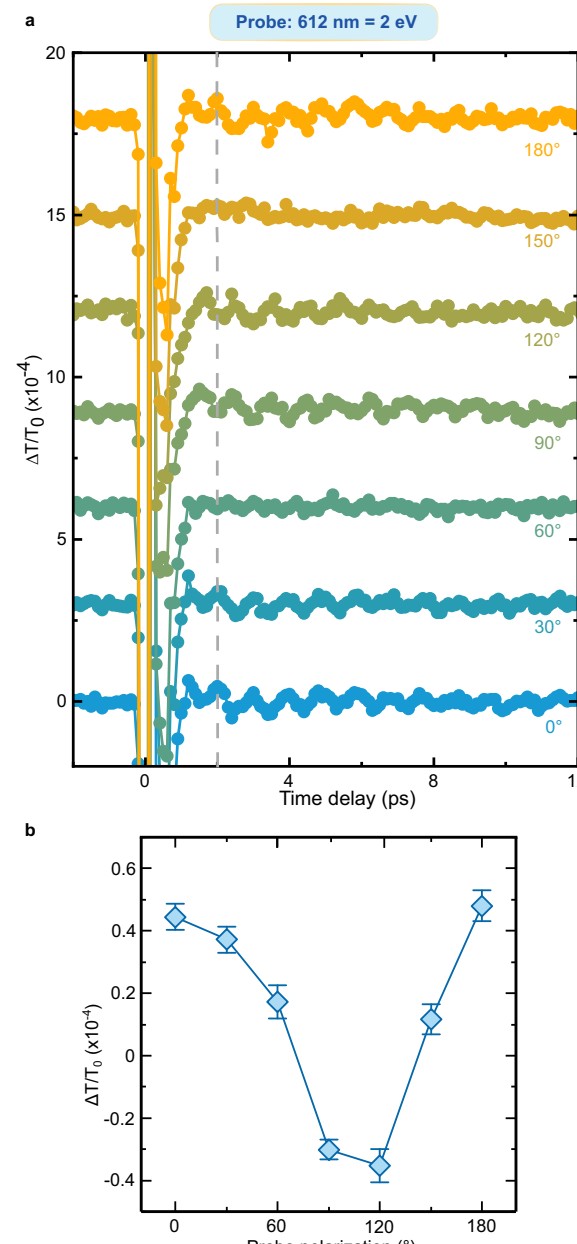

**Fig. 5 | Polarization dependence of the probe beam with 2.6 eV photon energy.** **a** Transient transmissivity for different orientations of the polarization of the probe pulses centered around 2.6 eV. The pump pulses are linearly p-polarized (0°, parallel to the z axis in Fig. 1c), with a fluence of 24 mJ/cm$^2$ and a central photon energy of 0.98 eV. The temperature of the NiO sample is set to 10 K. The dashed line highlights the change of sign of the amplitude within the dataset. **b** Dependence of the THz oscillation amplitudes on the orientation of the probe beam polarization. The amplitudes were obtained fitting the data in (**a**) with the function shown in Eq. (M1). The error bars are given by the standard error of the amplitudes obtained by the fitting the data in the time-domain. The standard error is calculated from the width of 95% confidence interval.

**Fig. 6 | Polarization dependence of the probe beam with 2 eV photon energy.** **a** Transient transmissivity for different orientations of the polarization plane of the linearly polarized probe pulses centered around 2 eV. The pump pulses are linearly p-polarized (0°, parallel to the z axis in Fig. 1c), with a fluence of 24 mJ/cm$^2$ and a central photon energy of 0.98 eV. The temperature of the NiO sample is set to 10 K. The dashed line highlights the change of sign of the amplitude within the dataset. **b** Dependence of the THz oscillation amplitudes on the orientation of the probe beam polarization. The amplitudes were obtained fitting the data in (**a**) with the function shown in Eq. (M1). The error bars are given by the standard error of the amplitudes obtained by the fitting the data in the time-domain. The standard error is calculated from the width of 95% confidence interval.

nonlinear spin dynamics into nonlinear electronic signals, which is essential for computing and logic gates.

## Methods
### Sample and characterization
Our sample of NiO is a single crystal with thickness of approximately 50 μm, cut along the 111 plane. The sample was commercially purchased. Figure 1b shows the measured transmissivity depending in the visible and near-infrared region. The temperature of the crystal was set to 10 K by means of a liquid-helium-flow cryostat. The light source for this experiment is an unpolarized tungsten white-light lamp. The radiation transmitted through the sample is detected by a Silicon spectrometer, granting a spectral resolution of 0.53 nm.

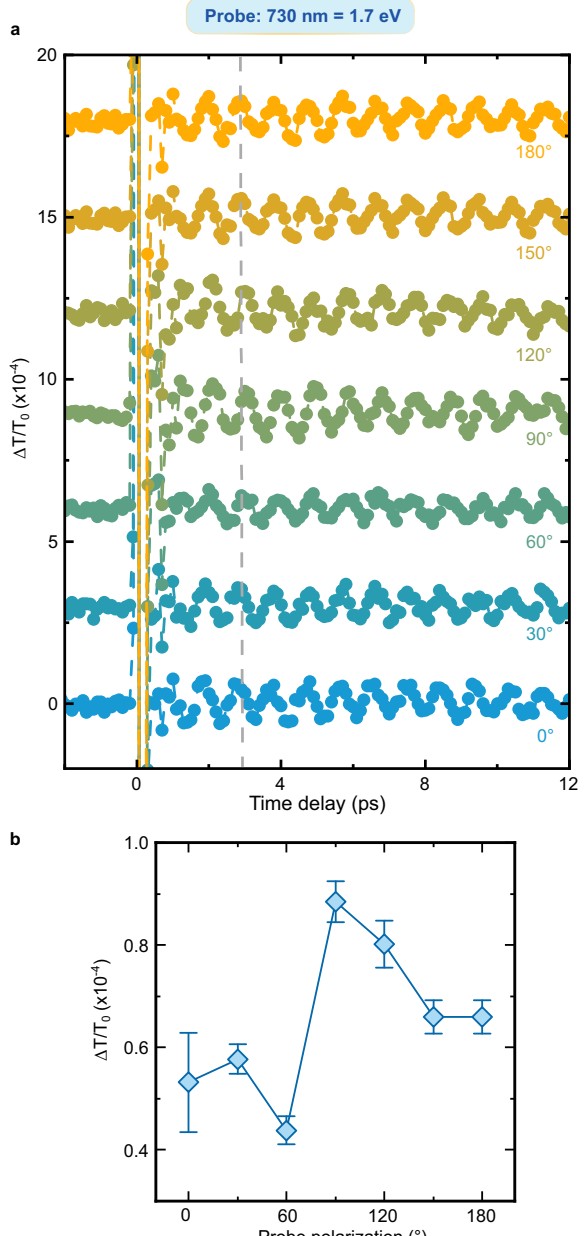

**Fig. 7 | Polarization dependence of the probe beam with 1.7 eV photon energy.**
**a** Transient transmissivity for different orientations of the polarization plane of the linearly polarized probe pulses centered around 1.7 eV. The pump pulses are linearly p-polarized (0°, parallel to the z axis in Fig. 1c), with a fluence of 24 mJ/cm² and a central photon energy of 0.98 eV. The temperature of the NiO sample is set to 10 K. The dashed line highlights the absence of change of sign of the amplitude within the dataset. **b** Dependence of the THz oscillation amplitudes on the orientation of the probe beam polarization. The amplitudes were obtained fitting the data in (**a**) with the function shown in Eq. (M1). The error bars are given by the standard error of the amplitudes obtained by the fitting the data in the time-domain. The standard error is calculated from the width of 95% confidence interval.

## Experimental set-up for magneto-optical pump-probe experiments

The pump-probe measurements are performed in the scheme shown in Fig. 1c. The laser source is an Yb:KGW amplified system emitting laser pulses with a central wavelength of 1026 nm and pulse energies of 400 μJ at a repetition rate of 50 kHz. The main output of the amplified system is split to pump a commercial and a home-made optical-

parametric amplifier (OPA). The home-made OPA is pumped by a residual of the fundamental output of the laser system with energy of 120 μJ. This OPA generates the pump beam spanning the spectral range from 0.89 eV to 1.13 eV. Coherent magnons are photoinduced by resonantly driving an exciton-magnon transition, with demands to tune the pump central photon energy to 0.98 eV. The probe beam is generated by the commercial OPA, whose output can be tuned in the 0.5–3.9 eV range. The duration of the pump pulses is on the order of 100 fs, while the duration of the probe pulses varies from 40 fs to 125 fs, depending on the central photon energy. The amplitude of the pump beam is modulated by a mechanical chopper, allowing the detection via a digital lock-in. We detect the transient transmissivity in a balanced-detection scheme (Fig. 1c). A beam splitter reflects a part of the probe beam before the interaction with the sample. This beam and the beam transmitted through the sample impinge on two photo-diodes. The difference of the voltages detected by the two diodes is tracked by our electronics and expresses the transient transmissivity. This scheme increases the sensitivity of our set-up by more than one order of magnitude, in comparison with the standard set-up relying on a single diode. Considering these two schemes, the balance detection enables us to improve the sensitivity from $\mathbf{8 \cdot 10^{-5}}$ to $\mathbf{3 \cdot 10^{-6}}$. The balanced detection eliminates noise sources due to fluctuations of the laser intensity and of the experimental conditions.

The global transmission $T$, i.e. not the photoinduced component at the frequency of the modulation introduced by the chopper, can be measured during the experiment, since our detection can discriminate each single laser pulse. With this value we can estimate the normalized transient transmissivity $\Delta T/T$.

### Data analysis
The pump-probe data were analysed using a time domain fit with the following equation

$$F(t) = a + b\,e^{-\frac{t}{\tau}} + a_{\text{GHz}}e^{-\frac{t}{\tau_{\text{GHz}}}} \cdot \cos\left(2\pi f_{\text{GHz}} \cdot t + \varphi_{\text{GHz}}\right) +$$

$$a_{\text{THz}}e^{-\frac{t}{\tau_{\text{THz}}}} \cdot \sin\left(2\pi f_{\text{THz}} \cdot t + \varphi_{\text{THz}}\right), \qquad \text{(M1)}$$

after the temporal overlap of pump and probe beam. The first two terms describe the background consisting out of a constant value $a$ and an exponential decay with a decay time $\tau$. The oscillations of both magnon modes of NiO are represented by the two decaying oscillations. The prefactors $a_{\text{GHz}}$ and $a_{\text{THz}}$ stand for the amplitudes of both oscillations, $\tau_{\text{GHz}}$ and $\tau_{\text{THz}}$ are the lifetimes, $f_{\text{GHz}}$ and $f_{\text{THz}}$ are the frequencies and $\varphi_{\text{GHz}}$ and $\varphi_{\text{THz}}$ are the phases of both oscillations.

Figures 2a and Fig. 3a present only the high frequency oscillation. This is achieved by the subtraction of a fit including the first three terms of Eq. M1. The presented amplitudes in Fig. 10d correspond to the fit parameter $a_{\text{THz}}$ of Eq. M1. The uncertainties are given by the standard errors resulting from the fit. An example of our fitting procedure applied to the original transient transmissivity (i.e. not only the THz component) is shown in Supplementary Fig. 2.

The transmissivity changes depicted in blue in Fig. 10d are obtained considering the energy shifts of the d-d transitions calculated with our model. For each probe wavelength employed in the experiment, we consider the bandwidth of the laser pulses. Then the energy shifts of all the d-d transitions, which lie within the relevant bandwidth of the pulses, are summed. We refer to this global energy shift with $\Delta E$. Next, we calculate the transmitted intensity $I_T$ of the probe beam, taking into account the spectrum of our sample (Fig. 1b) shifted by $+\Delta E$ or by $-\Delta E$. The two transmitted spectra, i.e. $I_T(+\Delta E)$ and $I_T(-\Delta E)$, are integrated over the bandwidth of the laser pulses. The difference between these two integrated quantities corresponds to the difference between maximum and minimum of the detected THz oscillations of $\Delta T/T$, namely the amplitude of the THz magnon mode.

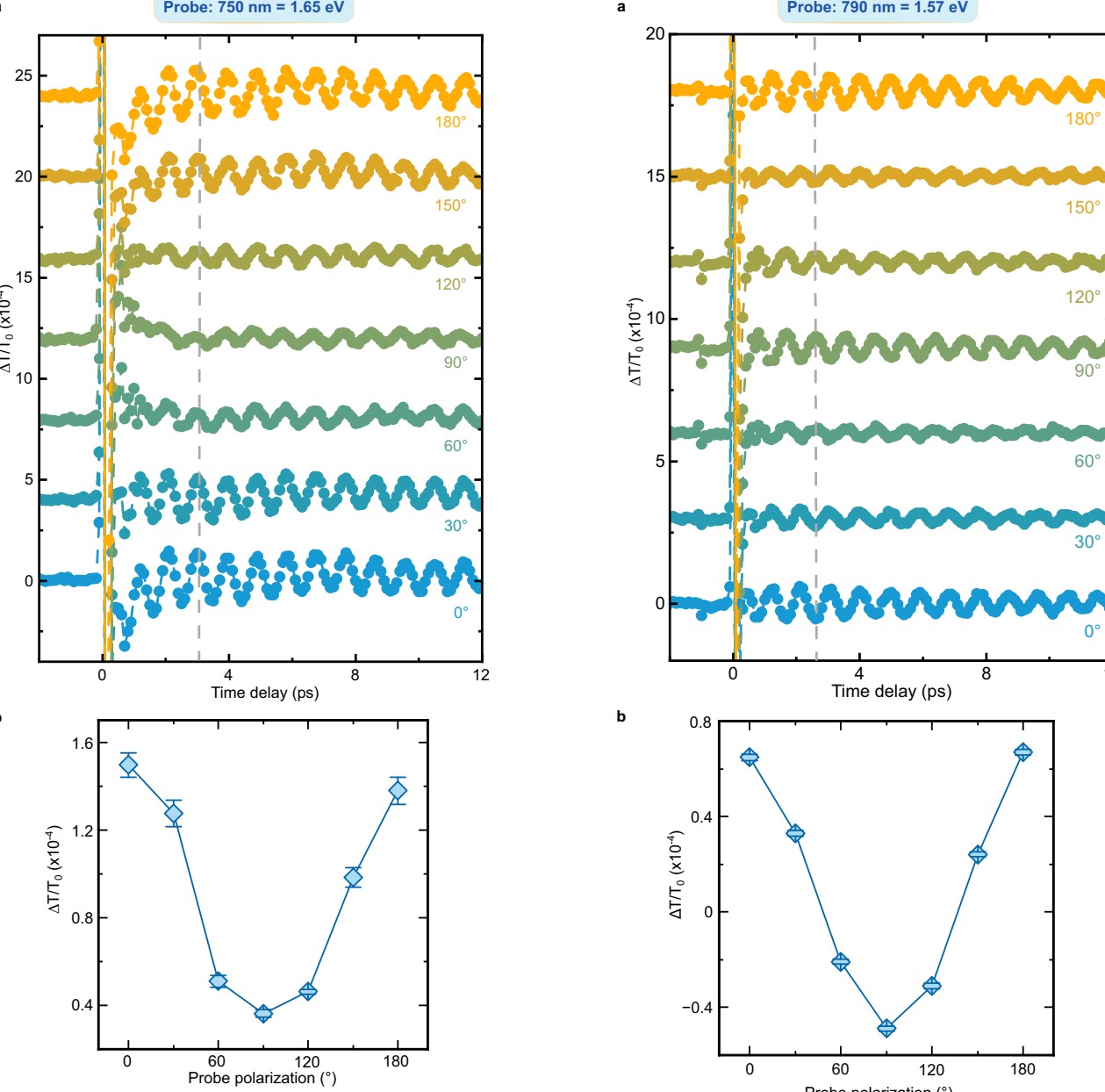

**Fig. 8 | Polarization dependence of the probe beam with 1.65 eV photon energy.**
**a** Transient transmissivity for different orientations of the polarization plane of the linearly polarized probe pulses centered around 1.65 eV. The pump pulses are linearly p-polarized (0°, parallel to the z axis in Fig. 1c), with a fluence of 24 mJ/cm² and a central photon energy of 0.98 eV. The temperature of the NiO sample is set to 10 K. The dashed line highlights the absence of change of sign of the amplitude within the dataset. **b** Dependence of the THz oscillation amplitudes on the orientation of the probe beam polarization. The amplitudes were obtained fitting the data in (**a**) with the function shown in Eq. (M1). The error bars are given by the standard error of the amplitudes obtained by the fitting the data in the time-domain. The standard error is calculated from the width of 95% confidence interval.

**Fig. 9 | Polarization dependence of the probe beam with 1.57 eV photon energy.**
**a** Transient transmissivity for different orientations of the polarization plane of the linearly polarized probe pulses centered around 1.57 eV. The pump pulses are linearly p-polarized (0°, parallel to the z axis in Fig. 1c), with a fluence of 24 mJ/cm² and a central photon energy of 0.98 eV. The temperature of the NiO sample is set to 10 K. The dashed line highlights the change of sign of the amplitude within the dataset. **b** Dependence of the THz oscillation amplitudes on the orientation of the probe beam polarization. The amplitudes were obtained fitting the data in (**a**) with the function shown in Eq. (M1). The error bars are given by the standard error of the amplitudes obtained by the fitting the data in the time-domain. The standard error is calculated from the width of 95% confidence interval.

## Magneto-optical effects and transmissivity

We analyze here the contribution of the four main magneto-optical effects on the optical transmissivity.

- *Faraday effect:* It is circular magnetic birefringence. As such, it changes the polarization of a linearly polarized beam propagating through the sample, by rotating it. It cannot affect the intensity of a transmitted beam. It is described by real anti-symmetric

components of the dielectric tensor, so it cannot take into account dissipative light-matter interaction, i.e. absorption.

- *Cotton-Mouton effect:* It is linear magnetic birefringence. Thus, it changes the polarization of linearly polarized beam propagating through the sample, by making it elliptical. It cannot affect the intensity of a transmitted beam. It is described by real symmetric components of the dielectric tensor, so it cannot

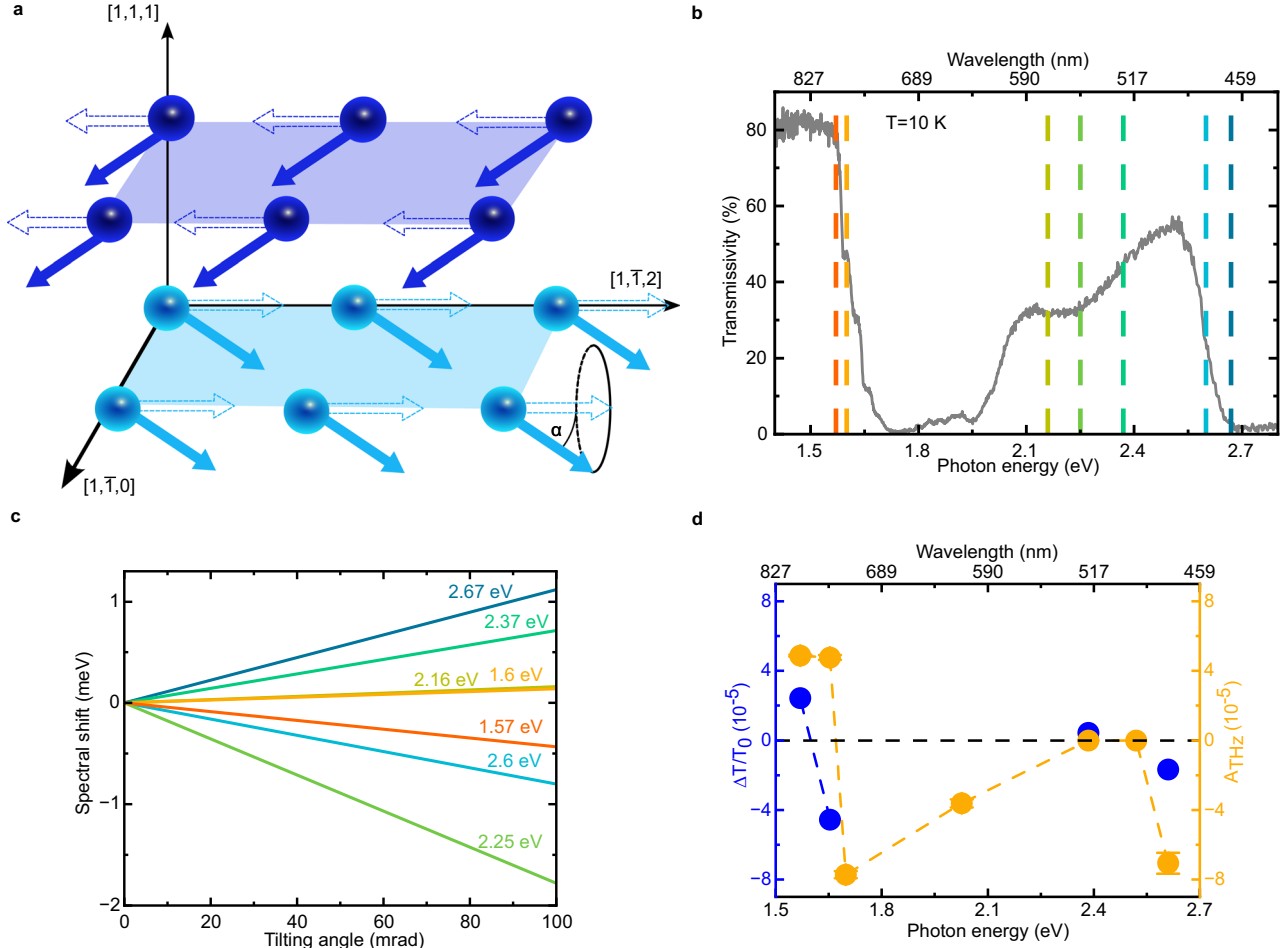

**Fig. 10 | Theoretical model and comparison of experiment and theory.**
**a** Scheme of the spins of the Ni²⁺ ions. The dashed arrows show the spins in the ground state, while the coloured arrows show the oscillating spins. The angle **α** parametrizes the spin deviation due to the magnon generation. **b** Static transmissivity of NiO at 10 K (grey line). The coloured dashed lines show the calculated energies of the electronic transitions of the Ni²⁺ ions. **c** Calculated spectral shifts of the electronic transitions depicted in (**b**) as a function of the tilting angle α.

**d** Comparison between the experimental values of amplitude of the THz magnon mode (orange) for different probe photon energies (Fig. 3a) with the calculated changes of the transmissivity (blue). The experimental values were obtained by fitting the time-traces in Fig. 3a with the last term of Eq. (M1). The angle α was set equal to 20 mrad. The error bars are given by the standard error of the amplitudes obtained by the fitting the data in the time-domain. The standard error is calculated from the width of 95% confidence interval.

take into account dissipative light-matter interaction, i.e. absorption.

- *Magnetic circular dichroism:* A linearly polarized beam can be decomposed into the sum of two circularly polarized beams (see Fig. 4a). If the linearly polarized beam experiences magnetic circular dichroism while propagating through a medium, the two circular components are not-equivalently absorbed. As a result, the intensity of the linearly polarized beam is reduced. This effect is independent of the linear polarization state of the beam in the x-y plane.

- *Magnetic linear dichroism:* This effect consists in a non-equivalent absorption of two components of a linearly polarized beam. For instance, in the figure below the absorption of the y-component is much more pronounced than the absorption of the x-component. Let's discuss first the idealized case of 100% absorption of the y-component and 0% absorption of the x-component (see Fig. 4b). If we now consider different linear polarization states (i.e. different values of the angle φ in Fig. 4b), we obtain the maximum of the transmissivity for φ = 0 (light parallel to the x-axis), while no transmissivity for φ = 90° (light parallel to the y-axis). This trend is periodic and it is described by the function 1 + cos(2φ), represented by the blue line in Fig. 4c. However, in a realistic material the transmissivity reaches neither 0% nor 100% values

along the two main axes, but values in between. It follows that magnetic linear dichroism in a real material affects the transmissivity according to A(1 + cos((2φ))}, with A < 1, taking values in the blue-box shown in Fig. 4c. Therefore, magnetic linear dichroism reduces periodically the amplitude of the transmissivity but cannot induce a change of sign.

All datasets displaying the probe polarization dependence obtained for different photon energies (except for the 1.65 eV measurements) cannot be interpreted in terms of magneto-optical effects (Figs. 5–9). In fact, they show either a change of sign in the transmissivity amplitude, or a trend that does not fit any predictions based on magneto-optical effects (1.7 eV data, Fig. 7). Although we cannot rule out a contribution from these effects, magneto-optics cannot be the only origin of the signal measured in all our datasets.

## Model
We compute the *d-d* transition energies by exact diagonalization of a Hamiltonian that contains the Coulomb-interaction, the crystal-field splitting, the spin-orbit coupling and a magnetic exchange field exerted by the adjacent ions. The Hamiltonian for the Coulomb interaction is given in Eqs. (98)-(101) of ref. 38. We have employed the spherical approximation which allows to express all Coulomb-interaction

parameters by the three Racah parameters, see Appendix C of Ref. 38. The Racah parameters were chosen as $A = 6.6$ eV, $B = 0.13$ eV, $C = 0.6$ eV[39]. These parameters, like the others, cannot be precisely determined. We have therefore used values found in the literature and have not attempted to achieve the best possible agreement with the experiment by fine-tuning these parameters. The other parameters were the crystal field $cf = 1$ eV[40], the spin-orbit coupling $s = 30$ meV[41], and the effective exchange field $E_{ex} = 60$ meV[42]. We neglect electronic band effects because the photon energies employed in our experiment, and considered in the model, lie below the fundamental absorption edge. In the ground state the effective magnetic field aligning the spins points in the $\langle 11\bar{2} \rangle$ direction, while it is tilted by the angle $\alpha$ because of the optical pumping of coherent zone center magnons. The tilted magnon state $|\alpha\rangle$ results from the magnon ground state by a global rotation

$$|\alpha\rangle = \exp\left(i\alpha \sum_j S_j^\perp\right)|0\rangle$$

$$= \exp\left(i\alpha \frac{1}{2} \sum_j \left(b_j^\dagger + b_j\right)\right)|0\rangle$$

and the population is computed after the Bogoliubov diagonalization to the diagonal operators $\beta_j^\dagger, \beta_j$ by the Baker-Campbell-Hausdorff formula

$$\frac{1}{N}\left\langle \alpha \left| \sum_j \beta_j^\dagger \beta_j \right| \alpha \right\rangle =$$

$$= \frac{\alpha^2}{4}\sqrt{\frac{1-\chi}{1+\chi}}$$

Here $S_j^\perp$ is a spin operator at Ni site $j$ perpendicular to the ground state magnetization, $b_j^\dagger$, $b_j$ are the local magnon creation (annihilation) operators before diagonalization, $\beta_j^\dagger, \beta_j$ the diagonalized magnon operators, $N$ the total number of lattice sites, and $\chi < 1$ the spin anisotropy factor, which we only introduce here to account for a small finite gap of the magnons.

## Data availability
Source data are provided with the article. Source data are provided with this paper.

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

## Acknowledgements

This work was supported by the Deutsche Forschungsgemeinschaft (DFG). D.B. acknowledges the support of the DFG program BO 5074/1-1; GSU acknowledges support by the DFG through UH 90/14-2. T.S. acknowledges the support of MEXT X-NICS (No. JPJ011438), NINS OML Project (No. OML012301), JST CREST (No. JPMJCR24R5), and JST ERATO (No. JPMJER2503). The authors thank Stephan Eggert and Christian Beschle for technical support.

## Author contributions

D.B. conceived and coordinated the project. M.C. developed the experimental setup, with contributions from V.W. and Ju. B. M. C. performed the measurements and analysed the results under the supervision of D.B. T.S. contributed to the symmetry analysis. G.S.U. conceived the model and the required calculations. J.B. provided the parameters and performed the calculations. All authors contributed to the interpretation of the results and the writing of the manuscript.

## Funding

## Competing interests

The authors declare no competing interests.
