## [Transparent Peer Review file · Nature Communications]

Coherence transfer from optically induced THz magnons to charges

Corresponding Author: Dr Davide Bossini

Version 0:

Reviewer comments:

Reviewer #2

(Remarks to the Author)

I think the authors have addressed my previous concerns properly, and the paper has been revised accordingly. I am happy to recommend this paper can be accepted for publication in Nat Commun.

Reviewer #3

(Remarks to the Author)

I would like to express my gratitude to the authors for their comprehensive responses to all my concerns, with which I am fully satisfied and convinced. The manuscript presents novel research on terahertz magnon mode-induced electric polarization control in dielectric antiferromagnetic NiO, evidenced as modulation in the crystal's transient transmissivity. Given the innovative nature of this work, I recommend it for publication in Nature Communications.

The revised paper provides clear and compelling evidence based on robust experimental results. However, I have identified a potential issue in Figure 3 of the revised version: there appears to be a discrepancy between the color-coded energies of time-delay scans in Figure 3a and the probe beam spectra in Figure 3b.

Overall, the findings in this study offer significant insights and contribute valuable knowledge to the field.

Reviewer comments:

Reviewer #2 (Remarks to the Author):

Comment 1

I think the authors have addressed my previous concerns properly, and the paper has been revised accordingly. I am happy to recommend this paper can be accepted for publication in Nat Commun.

Reply to comment 1

We thank the reviewer for the positive recommendation of our work and for their feedback, which helped us to improve the quality of our manuscript.

Reviewer #3 (Remarks to the Author):

Comment 1

I would like to express my gratitude to the authors for their comprehensive responses to all my concerns, with which I am fully satisfied and convinced. The manuscript presents novel research on terahertz magnon mode-induced electric polarization control in dielectric antiferromagnetic NiO, evidenced as modulation in the crystal's transient transmissivity. Given the innovative nature of this work, I recommend it for publication in Nature Communications.

Reply to comment 1

We thank the reviewer for the positive recommendation of our work and for their feedback, which helped us to improve the quality of our manuscript.

Comment 2

The revised paper provides clear and compelling evidence based on robust experimental results. However, I have identified a potential issue in Figure 3 of the revised version: there appears to be a discrepancy between the color-coded energies of time-delay scans in Figure 3a and the probe beam spectra in Figure 3b.

Reply to comment 2

We thank the Reviewer for noticing this problematic aspect of Figure 3a and 3b.

Action taken

We have modified the color scheme employed in the two panels to make them consistent with each other.

Comment 3

Overall, the findings in this study offer significant insights and contribute valuable knowledge to the field.

Reply to comment 3

We thank the Reviewer for this evaluation of the impact of our work.